# Amplification of electromagnetic fields by a rotating body

M. C. Braidotti ®[1], A. Vinante ®[2], M. Cromb ®[3], A. Sandakumar ®[3], D. Faccio ®[1,4] & H. Ulbricht ®[3] ✉

In 1971, Zel'dovich predicted the amplification of electromagnetic (EM) waves scattered by a rotating metallic cylinder, gaining mechanical rotational energy from the body. This phenomenon was believed to be unobservable with electromagnetic fields due to technological difficulties in meeting the condition of amplification that is, the cylinder must rotate faster than the frequency of the incoming radiation. Here, we measure the amplification of an electromagnetic field, generated by a toroid LC-circuit, scattered by an aluminium cylinder spinning in the toroid gap. We show that when the Zel'dovich condition is met, the resistance induced by the cylinder becomes negative implying amplification of the incoming EM fields. These results reveal the connection between the concept of induction generators and the physics of this fundamental physics effect and open new prospects towards testing the Zel'dovich mechanism in the quantum regime, as well as related quantum friction effects.

Electromagnetic (EM) wave amplification from a rotating cylinder was predicted by Yakov Zel'dovich in 1971[1]. His proposal is illustrated in Fig. 1a - a wave with angular momentum reflecting off a rotating and absorbing (e.g., metallic) cylinder will be amplified if the rotational Doppler shifted frequency of the incoming wave becomes negative (in the frame of the rotating cylinder)[1–3]. Negative frequencies or energies in a rotating system had already been pointed to lead to amplification by Penrose in the context of rotating black holes: particles falling into a black hole will acquire a negative energy as they pass through the ergosphere (point at which the spacetime drag velocity becomes larger than the speed of light)[4]. Penrose's reasoning points out that if the particle or mass splits so that part of the mass escapes or does not fall in, then this must gain energy in order to compensate for the negative energy of the part that falls into the black hole. In Zel'dovich's proposal, the black hole is replaced by a rotating cylinder but this does need not rotate faster than the speed of light. Rather, through purely classical calculations based on Maxwell's equations, Zel'dovich predicted the amplification of EM waves with frequency $\omega$ and angular momentum $\ell$ when

$$\omega - \ell\Omega < 0. \tag{1}$$

where $\Omega$ is the cylinder rotation frequency. This is the condition required for the Doppler shift (in the cylinder frame) to take the incoming wave frequency $\omega$ to negative values. When this condition is satisfied, the absorption coefficient changes sign, and the rotating medium loses part of its rotational energy to the outgoing waves, which are amplified[1–3].

Based simply on the existence of this *classical* amplification effect, Zel'dovich extrapolated that it should also be possible to amplify quantum fluctuations and therefore spontaneously generate EM waves at the expense of the cylinder rotational energy[1].

There have been many practical proposals to verify Zel'dovich's predictions in the classical regime[5–8], relying, e.g., on use of a very large orbital angular momentum (OAM) $\ell = 10,000$ or optically levitated particles spinning at GHz rates[7], restricting the size of the system to the nano-scale[9,10] or on synthetic approaches[8]. Despite the large quantity of proposals, technological challenges in meeting the amplification condition have prevented its verification with EM waves. The fastest rotation achievable by standard motors is of the order of 10 kHz (See for instance products of Celeroton, https://www.celeroton.com/en/products/motors/), and a record of 667 kHz is reported for a millimetre-sized magnetically levitated sphere[11]. The only

[1]School of Physics and Astronomy, University of Glasgow, G12 8QQ Glasgow, UK. [2]Istituto di Fotonica e Nanotecnologie - CNR and Fondazione Bruno Kessler, I-38123 Povo, Trento, Italy. [3]School of Physics and Astronomy, University of Southampton, SO17 1BJ Southampton, UK. [4]Institute of Photonics and Quantum Sciences, Heriot-Watt University, EH14 4AS Edinburgh, UK. ✉e-mail: h.ulbricht@soton.ac.uk

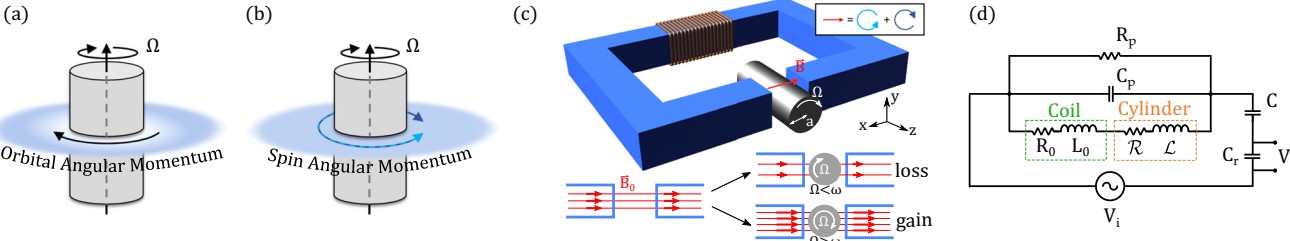

**Fig. 1 | Experiment concept and layout. a** Sketch of the 1971 Zel'dovich proposal: a doughnut-shaped wave with orbital angular momentum impinges on a rotating metallic cylinder. When condition Eq. (1) is met, the wave gains rather than loses energy due to the presence of the cylinder. **b** Sketch of our proposal: an EM field with spin angular momentum impinges on the rotating metallic cylinder. Both (co and counter-rotating) spin components arise from the decomposition of linearly polarised magnetic field **B₀** (blue and cyan arrows), as shown also in the inset of (**c**). **c** Experimental layout showing the rotating metallic cylinder placed in the gap of a toroidal ferrite. The coil encompasses a section of the toroid and induces an oscillating magnetic field **B₀** in the empty gap. The lower diagram shows in simplified terms how when the cylinder is present, according to Zel'dovich's theory, depending on its rotation speed Ω compared to the field oscillation frequency ω, it can either absorb or amplify (respectively reducing or increasing the total field **B** with respect to **B₀**). **d** Outline of the equivalent RLC circuit. $R_0$ is the ohmic coil dissipation, while dielectric losses are represented by $R_p$. The effect of the cylinder's speed on the field, and so the current and voltage in the circuit, is modelled as an effective resistance $\mathcal{R}$ (and inductance $\mathcal{L}$). Measurements are done over a readout capacitor $C_r$.

measurement of an analogue effect to date has been in the acoustic regime where the speed of light is replaced by the speed of sound that is orders of magnitude lower[12]. The key step forward in this acoustic work was the implementation of a scheme where the spinning disk is located deeply in the near-field regime, i.e., all dimensions (size of the OAM beam, radius, and thickness of the spinning disk) are much smaller than the field wavelength, $\ll \lambda$. In spite of this achievement, an EM version still remains an outstanding challenge and of interest due to the fundamental physical difference between mechanical rotation and EM fields (as opposed to acoustic waves that are also mechanical).

Here, we show that this 60-year-old long-sought effect has been concealed for all this time in the physics of induction generators. Induction motors are constituted of two components: an external stator, composed of circuits generating a rotating magnetic field, and a rotor, also composed of several elementary circuit loops, usually in a squirrel cage configuration. By replacing the internal circuits of the rotor with a solid metal cylinder as in Zel'dovich's original proposal, and using a gapped toroid within a LC resonator as stator, we isolate the key physical effect and unambiguously observe Zel'dovich amplification, which manifests itself as a negative dissipation induced by the rotor in the LC circuit. In this proposed experiment, a linearly oscillating magnetic field is generated in the toroid gap. The linearly polarised field can be decomposed into a superposition of equal co- and counter-rotating spin angular momentum components (Fig. 1b, c). The rotating absorbing metal cylinder placed in the gap field will 'see' these components with opposite rotational Doppler shifts. When the Zel'dovich condition (Eq. (1)) is met, the co-rotating component then gains energy from (rather than loses energy to) the cylinder; it is amplified with respect to the no-cylinder case. This experimental proposal overcomes two conceptual and technological difficulties related to high rotational speeds and low amplification, that were hindering the observation of this effect in the past: (1) having a near-field interaction allows us to use longer EM wavelengths; (2) using spin-rotational momentum, and not OAM, allows us to maximise the geometrical interaction area, by removing the non-wave zone intrinsic to OAM waves (see Fig. 1a, b). Importantly, by exploiting the mechanical equivalence between spin and OAM for the Doppler shift of EM waves[13,14], we overcome a key disadvantage pointed out by Zel'dovich himself: with OAM, the near-field interaction is necessary to avoid superluminal tangential velocities but leads to a very weak amplification. Indeed, the EM field amplitude decreases like a power of $(r/\lambda)^\ell$, thus leading to a weak or close-to-zero intensity in the region occupied by the rotating cylinder (see Fig. 1a) and consequent weak or negligible amplification[1,3].

## Results

To compute the Zel'dovich effect, we consider the equivalent circuit shown in Fig. 1d: the resistance $\mathcal{R}$ and the inductance $\mathcal{L}$ describe the effect induced by the rotating cylinder into the LC toroid circuit (see Methods Section for more details). When the condition Eq. (1) is satisfied for the co-rotating component, we expect the resistance $\mathcal{R}$ to become negative witnessing Zel'dovich amplification of the EM mode. Our best experimental estimations of the resistance and inductance induced by the rotor into the coil, $\mathcal{R} = R - R_0$ and $\mathcal{L} = L - L_0$, are shown in Fig. 2 as functions of the rotational frequency $F$ for the 4 different resonance frequencies $f_0$ of the LC oscillator, corresponding to different rotational frequencies of the magnetic field **B** in the gap. The continuous lines in the plots represent the theoretical curves for a rotating sphere. No free parameters were used to fit the model to the measurements, showing a substantial agreement with the experimental data. We note that the sphere-based model appears to capture more accurately border effects compared to the infinite cylinder model (See details in the Supplemental Material for additional information on the theoretical model with a sphere and with a cylinder). A residual discrepancy is however expected. The larger discrepancy in the inductance data could be explained by parasitic coupling to the brush-less motor, which would be stronger at lower frequencies due to an increased penetration depth. In Fig. 3 we directly show the amplitude data corresponding to the case of resonance frequency $f_0 = 277$ Hz for some selected values of the rotational frequency $F$. This plot illustrates more directly the amplification effect induced by the rotor in the LC resonator.

## Discussion

According to Zel'dovich's original paper, amplification occurs due to two principal ingredients: (1) the Doppler shifted frequency in the rotating frame of the object has to become negative; (2) the imaginary (absorptive) part of the rotor material susceptibility changes sign due to the negative Doppler frequency, transforming losses into gain. Furthermore, the shape of the absorption-to-amplification response relates to the ratio of the cylinder radius to the effective penetration depth in the rotating frame. Figure 2 shows that the experimental results follow the Zel'dovich trend for the resistance, and hence absorption, as a function of the rotor frequency[15]. When the rotor frequency $\Omega/2\pi$ exceeds the LC resonant frequency $f_0$, the co-rotating Doppler shifted frequency $\omega_- = \omega - \Omega$ and the corresponding resistance term become negative, marking the inflection point in the resistance $\mathcal{R}$ plot. At slightly higher frequency the negative co-rotating

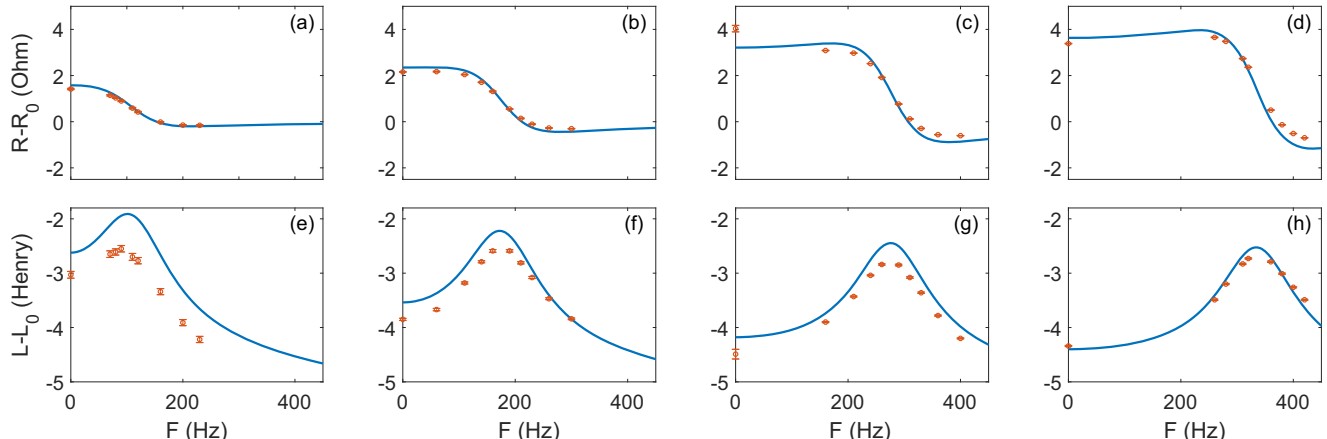

**Fig. 2 | Resistance and inductance induced by the rotor in the coil.** Resistance $\mathcal{R} = R - R_0$ (**a–d**) and inductance $\mathcal{L} = L - L_0$ (**e–h**), as function of the cylinder rotational frequency $F = \Omega/(2\pi)$ for different sets of capacitors, and hence different resonance frequencies of the LC resonator $f_0 = \omega/2\pi$. **a, e**: $C = 10$ nF, $C_r = 47$ nF, $f_0 = 107$ Hz; (**b, f**): $C = 3.3$ nF, $C_r = 22$ nF, $f_0 = 175$ Hz; (**c, g**): $C = 1.0$ nF, $C_r = 22$ nF, $f_0 = 277$ Hz; (**d, h**): $C = 1.0$ nF, $C_r = 1.0$ nF, $f_0 = 335$ Hz. The values obtained by the sphere model are shown as a reference (continuous blue lines). The vertical lines mark $f_0$. Error bars of individual values are barely visible at the plotted scales. We notice that the negative values of $\mathcal{R}$ are negative by 6 std. dev. at $f_0 = 107$ Hz and by 75 std. dev. at $f_0 = 335$ Hz.

term exceeds the positive counter-rotating one, leading to a negative total resistance $\mathcal{R}$. This is a signature that the absorption coefficient has flipped sign and hence that there is an electromagnetic gain induced by the mechanical rotation.

The effect is particularly evident at high resonance frequencies ($f_0 = 277$ and $335$ Hz) in agreement with the model, i.e., the imbalance between the negative resistance (gain) induced by the co-rotating component of the field and the positive resistance (absorption) induced by the counter-rotating component, increases with the magnetic field frequency $f$.

The fact that the negative resistance induced by the rotor corresponds to an effective amplification of the EM field is shown more directly in Fig. 3. Here, at low $F$ the peak amplitude at resonance is reduced with respect to the no-rotor case due to the eddy current dissipation in the rotor. However, as $F$ is increased to higher values the peak amplitude increases as well (as indicated by the arrow in Fig. 3). Eventually, beyond the Zel'dovich threshold, i.e., when the cylinder rotation $F$ is higher than the circuit resonance frequency $f_0$, the dissipation induced by the rotor flips sign leading to an amplification of the EM field and the peak amplitude at the resonance becomes larger than in the case with no rotor. As in Fig. 2, the presence of the counter-rotating spin component provides an effective loss term and causes the gain to appear at frequencies slightly above the predicted Zel'dovich threshold ($\Omega > \omega$).

We conclude that our experiment, based on the simplest possible interaction of a solid metallic cylinder with an oscillating magnetic field, is substantially reproducing the electromagnetic amplification mechanism predicted by Zel'dovich. Furthermore, it is making a further step showing that this effect can be generalised to spin angular momentum. The practical impossibility to test Zel'dovich's predictions, pointed out already in the original paper[1], is overcome here thanks to the heavy spatial confinement of the electromagnetic field in a LC resonator, compared to free space[15], and by making use of spin angular momentum.

At the same time, these results show an unexpected connection between the Zel'dovich effect and induction generators[16], which extract electric power from rotational motion. We identify the Doppler shifted frequency with the 'slip frequency' in induction motor terminology, and ingredient (1) is satisfied in the generator regime, when the rotor is driven faster than the rotating magnetic field induced by the stator excitation current. Our setup directly implements condition (2) with a homogeneous metallic rotor. However, typical induction

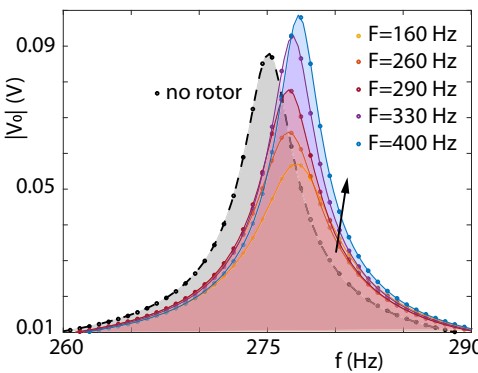

**Fig. 3 | Experimental measurement of the EM voltage amplitude.** Voltage $|V_o|$ measured as a function of the EM field frequency $f = \omega/(2\pi)$ at $f_0 = 277$ Hz (obtained with capacitors $C = 1.0$ nF, $C_r = 22$ nF). The dashed black curve shows the bare LC resonance with no metallic cylinder. The coloured curves show measurements for different cylinder mechanical frequencies $F = \Omega/(2\pi) = (160, 260, 290, 330, 400)$ Hz in colours (yellow, orange, red, purple, blue). The peak amplitude at the resonance at the lowest rotation frequency, $F = 160$ Hz is significantly lower than the no-rotor case due to eddy current losses in the rotor. As we increase the rotation rate, the peak amplitude increases and becomes larger than the no-rotor case (the arrow indicates the general increasing trend of the curves with increasing F). The shifts in resonant frequency are caused by the inductance induced by the rotor, consistently with Fig. 2g.

generators are made with a squirrel cage rotor, i.e., circuits formed by multiple conducting bars around an iron core. Rather than the response of the rotor being determined by a solid material susceptibility, it is engineered through the resistances and inductances of the squirrel cage circuits. The analogue susceptibility of a squirrel cage enhances the amplification, for a more efficient generator than Zel'dovich's original proposal.

It is worth finally commenting on the links between the Zel'dovich effect and other amplification mechanisms that could be extended to the quantum regime where one might expect the spontaneous generation of waves and the slowing down of the cylinder. This then provides a connection to the broader family of 'quantum friction', i.e., the slowing down of a moving or rotating body as a result of interaction with the quantum vacuum. For example, an oscillating cavity or body is predicted to lead to the amplification of vacuum fluctuations, often referred to as the dynamical Casimir effect[17–19]. This effect

however is fundamentally different from Zel'dovich's prediction. The dynamical Casimir effect leads to amplification at $\Omega/2$ ($\Omega$ here would be the frequency of oscillation/rotation of the boundary - cavity or body) and has no threshold, i.e., it occurs for all oscillation frequencies (the Zel'dovich effect has a clear threshold as shown in Eq. (1)); it is essentially a parametric oscillator or amplifier (the Zel'dovich effect arises from a change in sign of the absorption coefficient, passing therefore from exponential attenuation to exponential gain); it can in principle be observed with either absorbing or dielectric particles alike (the Zel'dovich effect relies quintessentially on the presence of absorption and losses); it requires some form of geometrical symmetry breaking, e.g., a smooth cylinder rotating along its longitudinal axis will not lead to DCE (but is the exact condition required by Zel'dovich). With these observations one can therefore appreciate the details and differences between proposals for the observation of quantum friction or amplification of EM waves through mechanisms akin to the dynamical Casimir effect (see e.g., refs. 20,21) that hold promise also in the optical domain, versus Zel'dovich amplification.

In summary, by operating in the sub-wavelength regime and using spin angular momentum, we have experimentally measured negative dissipation induced by a rotating metallic cylinder, indicating the amplification of EM waves originally predicted by Zel'dovich. This 60 year old prediction is found to be observable thanks to the unforeseen link between Zel'dovich amplification and induction motors. These findings open the way to the merging of ideas from two previously disconnected fields. In particular, a suggestive prospect is the realisation of Zel'dovich electromagnetic amplification from a rotating body in the quantum regime[5,22], i.e., the generation of photons out of the quantum vacuum stimulated by a mechanical rotation[23–25]. Induction motor schemes can be optimised through high efficiency magneto-mechanical coupling and could in the future be used in the quantum regime. It remains true that realistic values of $\omega, \Omega \sim 10^3$ Hz imply that the temperature required to bring the resonator in the ground state would be challengingly low $T \sim 10^{-9}$ K. However, ground-state cooling of a low frequency LC resonator can be rather achieved using techniques borrowed from optomechanics, such as feedback-cooling[26]. Such experiments would allow to observe Zel'dovich amplification in the quantum regime. Similarly, we expect that some of the approaches used here, e.g., resorting to low-frequency EM waves and the near-field configuration enabled by the interaction based on spin angular momentum, will be of use also for the further development and experimental implementation of other mechanical-to-EM wave transduction and amplification schemes.

## Methods

### Experimental setup

In our experimental setup the rotor is an aluminum cylinder with radius $a = 2$ cm, mounted on a brush-less motor[27] which can be spun up to 500 Hz about its symmetry axis (see Fig. 1c). The magnetic field is generated by a coil wound around a gapped toroidal ferrite core with square section $4 \times 4$ cm. The rotor is inserted in the 4.4 cm gap, slightly larger than rotor diameter. The coil is made of $2 \times 10^4$ turns of 0.2 mm diameter copper wire. The ohmic coil resistance at room temperature is $R_0 = 2.03$ kΩ, while the measured coil inductance is $L_0 \approx 263$ H. We approximate the magnetic field as quasi-uniform over the gap. The current-to-field factor $\beta = (0.40 \pm 0.03)$ T/A has been directly calibrated with a Gaussmeter (Hirst GM05) placed in the centre of the gap.

### The model

Our setup is composed of a LC toroid circuit and a spinning cylinder within the toroid gap, as shown schematically in Fig. 1c.

To compute the effect of the rotating cylinder on the circuit, we first calculate the currents induced by the magnetic field on the metallic cylinder, and then the field induced by the currents back into

the circuit. Following the calculation in ref. 15, we consider the spinning cylinder axis oriented along $z$ and the magnetic flux density $\boldsymbol{B_0}$ produced by the coil wrapped around the toroid to be uniform and oscillating at a frequency $\omega$. In the lab reference frame we can write the field in the toroid gap as

$$\mathbf{B_0} = \beta \mathbf{b_0} I, \tag{2}$$

where $I$ is the current in the coil wrapped around the toroid. The factor $\beta$ is a geometrical factor that we measure experimentally. The vector $\mathbf{b_0} = (1, 0, 0)^T e^{i\omega t}$ indicates the linear polarisation of the oscillating magnetic field. We write this as the sum of a co-rotating and a counter-rotating vector with respect to the cylinder axis, $\mathbf{b_0} = 1/2 \left[ (1, i, 0)^T + (1, -i, 0)^T \right] e^{i\omega t}$. We then move to the reference frame co-rotating with the cylinder, spinning at frequency $\Omega$:

$$\mathbf{b_0'} = \frac{1}{2} \left[ (1, i, 0)^T e^{i(\omega - \Omega)t} + (1, -i, 0)^T e^{i(\omega + \Omega)t} \right]. \tag{3}$$

The response of the cylinder to the applied magnetic field is thus the superposition of the response to the co-rotating and the counter-rotating polarisation components, which rotate at different Doppler shifted frequencies, $\omega_\pm = \omega - s\Omega$, where $s$ indicates the wave spin ($s = 1$ co-rotating, $s = -1$ counter-rotating). In particular, when the condition $\Omega > \omega$ is fulfilled, the co-rotating frequency ($\omega_- = \omega - \Omega$) flips sign, hence the Doppler shifted frequency of that incoming field becomes negative.

The response of the cylinder to each component can be found by solving Maxwell's equations for the specific geometry and material. Analytical solutions can be found in the case of a spherical rotor[15] and an infinite cylinder (see Supplemental Material for additional information on the theoretical model with a sphere and with a cylinder). For practical purposes, the sphere solution is found to adequately describe our experiment, as shown below. For our conductive rotor, the response is determined by the eddy currents induced in the conductive material of the cylinder, involving both an inductive (in-phase) and resistive (out-of-phase) component. Induced eddy currents will couple a magnetic flux back into the coil circuit, $\Phi(\omega) = \alpha(\omega \pm \Omega)I(\omega)$, where the linear response function $\alpha = \beta^2 \chi$ is evaluated in the rotating frame. Here, $\beta$ is the field geometrical coupling and $\chi = \chi' - i\chi''$ is the susceptibility, i.e., the complex response function of the cylinder to the rotating magnetic field. The components $\chi'$ and $\chi''$ are the in-phase and out-of-phase components of this response function, respectively. Moreover, according to linear response theory, $\chi(-\omega) = \chi^*(\omega)$, in particular $\chi''(-\omega) = -\chi''(\omega)$, where $\chi''$ is the absorption component. This odd-symmetry for negative frequencies is the key part of Zel'dovich's amplification prediction. Physically, the magnitude of the response with $\Omega$ is system-dependent and in general, it is not monotonic, i.e., the amplification gain does not simply increase with rotation $\Omega$. This can be understood to relate to the ratio of the cylinder radius to the penetration depth (at which eddy currents can circulate) in the rotating frame, which scales as $1/\sqrt{|\omega \pm \Omega|}$. When the radius and effective penetration depth are of similar size (such that the field is penetrating far into the cylinder, but not so far that it just passes through) there is maximum interaction. This in turn implies that for large rotation rates $\Omega \gg \omega$, the magnitude of the effect tends towards zero (the no-cylinder case), in keeping also with the theoretical predictions, e.g., in refs. 3,15. This behaviour can be clearly seen in Fig. 4 that shows an example calculation of the Zel'dovich gain for our system, as calculated using the theory described in this section (see also the Supplementary Material, for additional information on the theoretical model with a sphere and with a cylinder).

From an experimental point of view it is convenient to define the resistance $\mathcal{R}$ and the inductance $\mathcal{L}$ induced by the rotating cylinder

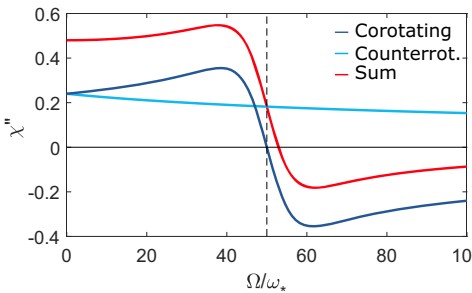

**Fig. 4 | Theoretical model.** Dissipative response of a spherical rotor $\chi''$ as function of normalised rotation frequency for the the two components corotating (blue line) and counterrotating (cyan line). The red line is the sum of the two terms, showing that the total dissipation can indeed become negative. Here the field frequency is set to $\omega = 50\omega_*$ (dashed vertical line), where $\omega_* = (\mu_0\mu_r\sigma a^2)^{-1}$ is a scaling frequency at which the penetration depth equals the radius $a$ of a spherical rotor with permeability $\mu_0\mu_r$, and conductivity $\sigma$ (see Supplemental Material for additional information on the theoretical model with a sphere and with a cylinder).

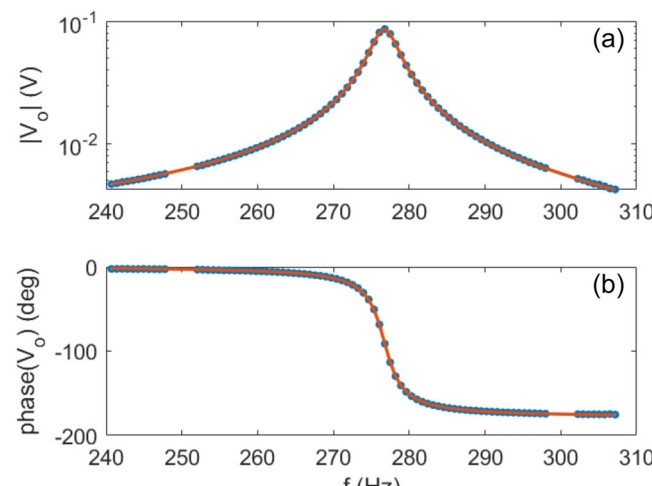

**Fig. 5 | Example of the amplitude response.** $|V_o|$ (**a**) and phase response $phase(V_o)$ (**b**) of the LC coil-capacitor setup, measured by the lock-in amplifier, as a function of the generator frequency $f = \omega/(2\pi)$. In this example the rotor mechanical frequency is fixed at $F = \Omega/(2\pi) = 330$ Hz and the LC resonance frequency is $f_0 = 277$ Hz, obtained with capacitors $C = 1.0$ nF, $C_r = 22$ nF. From the fit, shown as the red lines with modulus and phase of Eq. (6), we can extract the parameters $L$ and $R$.

into the circuit:

$$\mathcal{R} = Re[V/I] = \omega\beta^2[\chi''(\omega - \Omega) + \chi''(\omega + \Omega)] \quad (4)$$

$$\mathcal{L} = Re[\Phi/I] = \beta^2[\chi\prime(\omega - \Omega) + \chi\prime(\omega + \Omega)], \quad (5)$$

where $V = i\omega\Phi$ is the voltage induced into the circuit by the rotating cylinder.

The in-phase component $\beta^2\chi\prime$ gives the variation of inductance $\mathcal{L}$ generated by the presence of the rotating cylinder in the gap, while $\omega\beta^2\chi''$ can be seen as an induced resistance. Hence, the condition $\omega - \Omega < 0$ for the co-rotating component leads to a negative resistance that in turn implies a power emission into the EM mode as opposed to the expected (for a non-rotating or slowly rotating cylinder) power absorbed from the EM mode. This corresponds to the amplification predicted by Zeldovich in free space[1–3].

However, note that in the case of a linearly polarised oscillating field the resistance $\mathcal{R}$, induced by the presence of the cylinder, is always composed of a co-rotating and a counter-rotating component. We can have amplification only if the negative resistance induced by the co-rotating term is larger than the counter-rotating one (which is always positive). This can indeed occur due to the symmetry breaking induced by the mechanical rotation (as illustrated in Fig. 4) and the counter-rotating term could be completely eliminated in a setup with a circularly polarised field.

**Measurement and data analysis**

In order to perform accurate measurements of the resistance $\mathcal{R}$ and inductance $\mathcal{L}$ induced by the rotor, the coil is placed in series to two capacitors, $C$ and $C_r$, thus forming a *RLC* circuit, whose scheme is shown in Fig. 1d. The capacitor $C_p$ and the resistance $R_p$, in parallel to the coil + cylinder system, represent the parasitic capacitance and associated dissipation of the coil. The value $C_p = 310$ pF has been estimated in a separate measurement of the bare coil, without other capacitors connected, together with the coil ohmic resistance $R_0 = 2.03$ kΩ. The RLC circuit is powered by an input voltage $V_i$, applied by a function generator. Without this applied input voltage only noise was measured in the system and there was insufficient signal to noise ratio to measure any effect of the cylinder, which in this circuit only contributes a fraction of the overall resistance.

We measure the complex transfer function of the output voltage $V_o$ across the large readout capacitor $C_r$. The measurement is performed by a lock-in amplifier[28] synchronous with the input signal. The capacitor $C_r$ has an impedance much lower than the lock-in input impedance $R_{in} = 1$ MΩ, in order to suppress the parasitic dissipation induced by $R_{in}$.

A typical measurement of amplitude and phase of the output voltage $V_o$, measured by the lock-in amplifier, as functions of the generator frequency $f = \omega/2\pi$ is shown in Fig. 5. Amplitude and phase response of the RLC resonator are then fitted by modulus and phase of the complex transfer function of the circuit in Fig. 1d, which can be expressed as:

$$V_o = \frac{Z_r}{Z_r + Z_c + \frac{1}{\frac{1}{R_p} + i\omega C_p + \frac{1}{Z_l}}} V_i, \quad (6)$$

where $Z_r = 1/(i\omega C_r)$, $Z_c = 1/(i\omega C)$, $Z_l = R + i\omega L = R_0 + \mathcal{R} + i\omega(L_0 + \mathcal{L})$. $R = R_0 + \mathcal{R}$ and $L = L_0 + \mathcal{L}$ are the total resistance and inductance of the coil+cylinder system, respectively.

The resonant frequency of the circuit determined by Eq. (6) is approximately $f_0 = 1/2\pi\sqrt{LC_{eq}}$, where the equivalent capacitance is $C_{eq} \approx (C^{-1} + C_r^{-1})^{-1} + C_p$. Thus we can tune $f_0$ by varying $C$ and $C_r$.

We measured the amplification for 4 different resonance frequencies of the circuit $f_0$, engineered by 4 different sets of capacitors $C$, $C_r$. This allows us to test our model over a wide range of parameters. For each resonance frequency $f_0$, we first estimate the inductance $L_0 \simeq 263$ H and the parasitic resistance $R_p \simeq 60$ MΩ by fitting the amplitude and phase of the output voltage with the rotor removed from the gap. Then, we insert the rotor and vary the rotor frequency $F = \Omega/2\pi$ over a range including values lower and higher than $f_0$. For each $F$ we fit again the amplitude and phase response, but now all parameters $C, C_r, C_p, R_p$ are kept as fixed parameters, and only $R$ and $L$, embodying the effect of the cylinder, are left as free parameters. The values of $R$ and $L$ extracted from amplitude and phase fits are typically consistent with each other, with relative discrepancies of the order of $10^{-4}$ for $L$ and $10^{-2}$ for $R$.

## Data availability

Data is available at: https://doi.org/10.5525/gla.researchdata.1457

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

## Acknowledgements

We thank Damon Grimsey for expert technical support with the setup. The authors acknowledge financial support from EPSRC (UK Grant No. EP/P006078/2) (D.F.) and the European Union's Horizon 2020 research and innovation programme, grant agreement No. 820392 (D.F.). We further acknowledge financial support from the QuantERA grant LEMAQUME (A.V.), funded by the QuantERA II ERA-NET Cofund in Quantum Technologies implemented within the EU Horizon 2020 Programme, from the UK funding agency EPSRC (grants EP/W007444/1 (H.U. and D.F.), EP/V035975/1 (H.U.), and EP/X009491/1 (H.U.)), the Leverhulme Trust (RPG-2022-57) (H.U.), the EU Horizon 2020 FET-Open project TeQ (766900) (H.U.) and the EU Horizon Europe EIC Pathfinder project QuCoM (GA no.10032223) (H.U.).

## Author contributions

M.C.B. and A.V. conceived the experiment. H.U. built the experimental setup, and A.S., H.U., and M.C. took data. M.C.B., A.V., and M.C. analysed the data. All authors discussed the results and wrote the manuscript. H.U. and D.F. acquired the funding.

## Competing interests

The authors declare no competing interests.
