## [Peer Review File · Nature Communications]

Amplification of electromagnetic fields by a rotating bodyREVIEWER COMMENTS

Reviewer #1 (Remarks to the Author):

The authors report on an experiment demonstrating the amplification of an LC oscillating circuit by the mechanical rotation of a metallic cylinder, in close analogy with an effect predicted by Zeldovich more than 50 years ago. The setting differs appreciably from the one considered by Zeldovich, where radiation with orbital angular momentum is scattered off the cylinder, rather than the latter being immersed in an oscillating homogeneous magnetic field. Nonetheless, I agree with the authors that it is based on the same underlying physics. Incidentally, the experiment is conceptually and technically relatively simple, and could have probably been done 50 years ago.

This is a well written manuscript, clearly exposing the physics, the experimental methods and the theoretical analysis. Since this is a first demonstration of field amplification by a rotating object, I do recommend publication in Nature communications.

Remarks:

The abstract states that this is "a key piece of fundamental physics". This rather strong statement is not really justified by the discussion in the manuscript, in my view. I therefore suggest that the authors explain more clearly what they mean (or maybe tone down their wording).

The authors should carefully check their Supplemental Material for consistency, clarity, and correctness. For instance, the LHS of (3) needs a prime, ω_{pm} in (6) is not defined until the middle of page 3, the ϕ on the 2nd line of page 3 should probably be a Φ , what is σ in (9)?, the $f()$ in (15) is not clearly defined, the relation of F in (14),(15) to $F_{co}, F_{counter}$ is not clear, the F_{co} in (19) should probably be a $F_{counter}$, etc.

Reviewer #2 (Remarks to the Author):

The authors find a situation that mimics the amplification of electromagnetic waves (waves in general) by bodies rotating faster than the optical period, as predicted by Zeldovich. The paper is well-written and portrays an exciting configuration that relates to the original prediction. The connection is nice and worth being reported in a high-impact journal such as Nature Communications.

Before the paper is published, I would suggest that the authors produce a more tutorial description of the connection between their system and a rotating body amplifying some waves, as discussed by Zeldovich. As it stands, the paper is highly technical, so only understandable by an intersection of two communities (electrical engineering and electromagnetic theory).

In addition, I would suggest the authors discuss in more detail which aspects of the original prediction are covered here, as well as a better discussion of the different considerations that have

been considered in the literature. The list of references misses important contributions in the field that can transmit an impression biasing.

In particular, a rotating body in a vacuum in which light waves are far from any sources constitutes an appealing class of amplification and spontaneous emission systems driven by mechanical rotation. These aspects have been covered in the following papers that should be appropriately cited in the manuscript:

-- Manjavacas and Garcia de Abajo, Phys. Rev. Lett. 105, 113601 (2010)

-- Asenjo-Garcia et al., Phys. Rev. Lett. 106, 213601 (2011)

The spontaneous generation of radiation should occur for any rotation frequency in objects capable of absorbing at low frequencies (for example, dissipative metals). The process is associated with a simultaneous excitation in the material, emission of a photon, and change in the rotation state. The description in the manuscript of this point seems to be confusing and does not emphasize that this is a mechanism taking place even rotationally symmetric bodies.

The authors claim that it is impossible to observe amplification with light ("impossibility to test Zel'dovich predictions"), but some possible systems have been discussed in the literature including the Asenjo-Garcia et al. paper.

The system studied by the authors differs in substantial aspects from a situation in which one has electromagnetic wave propagating in a medium rather than resonant modes is a more or less quasistatic configuration.

The conclusions in the paper seem to transmit the impression that the present work closes the subject by demonstrated something that is possible (and that can be further studied in analogous systems), and that any other configuration (particle in vacuum) is impractical.

The similarities and differences with respect to the original proposal, and the range of possibilities should be discussed in a more balanced way.

These aspects should be commented in the manuscript.

Reviewer #3 (Remarks to the Author):

This work shows an experimental demonstration of amplification of radio waves via the wave scattering process with a rotating metallic cylinder. The physics root in the negative Doppler effect, where the gain is supposed to come from the mechanical rotational energy from the spinning cylinder. After going through the work, I cannot recommend its publication in NC, due to the lack of novelty in physics and tight limitations in applications. Also the amount of work of this paper is far short of the bar of NC.

The detailed reasons are listed as follows:

1. The physics was actually pointed out in many previous works, such as Refs 11-13. The difficulty in realizing the amplification effect is mainly technical. For example, it is very challenging to rotate a cylinder super fast up to the frequency of light in THz or even sound in the kHz.
2. In this work, the authors propose a LC circuit to generate radio waves with a very low frequency of hundreds of Hz which smartly relieve the tight requirement of rotation speed of cylinder. Such proposal cannot be easily extended into other fields such as acoustics and photonics.
3. This work has nothing to do with the quantum regime but only a LC circuit that generates radio waves of very low frequency and interacts with a spinning cylinder.

Reviewer #4 (Remarks to the Author):

Authors reports an experimental method to observe the Zel'dovich prediction of amplification of electromagnetic wave by an induction motor with a spinning metal rod. The topic is very interesting and potentially provides a novel way to verify a novel interaction between the mechanical spin and electromagnetic (EM) wave. The measuring method the authors figured out is very clever. However, as for me, what the authors measured is only the negative resistance in the rotator-LC resonator, but not the actual gain or amplification of EM wave. The measured negative resistance just confirms the fact that the mechanical energy from the spinning object can be transformed to EM energy, but not to EM wave as the authors claimed in the manuscript. Although the work is interesting, the physical process beneath the energy transform from the spin mechanics to EMs cannot be directly measured by the experiment. Thus, the "amplification of EM waves" in the title cannot accurately describe the actual work in the manuscript and may mislead reader. To confirm the conclusion more firmly and more rigorously, authors should present more measurements on the radiation of the EM wave when the spin speed is beyond the resonant frequency, confirming the amplification of the EM wave?

My other concern is about the style of writing the manuscript. In the manuscript, the main novel idea beneath the experimental method is somewhat abstract and physical to make some difficulty for readers. Could authors give some figures to explain the idea of the experiment and its connection to the Zel'dovich prediction?

My third concern is about the claim in the manuscript that the experiment could observe EM amplification in quantum regime. It is well known that the noise induced by quantum fluctuation in vacuum EM fluctuation may be far small than the noise in electronic system as used in the experiment even in the ultra-low environmental temperature. So, I think the claim is not rigorous and the EM amplification in quantum regime may not be observed.

As for me, the manuscript in current form is not suitable for publication on journal Nat. Comm. and may be more suitable for publication in a specialized journal.

Response to Reviewer #1– NCOMMS-23-60951-T

We would like to thank the reviewer for taking the time to review the manuscript, and for giving constructive feedback. We are pleased that the reviewer agrees our paper should be published in Nature Communications.

We have responded to all the comments in detail below. We have attached a copy of the manuscript with changes highlighted in red. Additionally, some wording in the manuscript has been tweaked to increase readability.

- Comment 1

1. **Reviewer Comment:** The abstract states that this is “a key piece of fundamental physics”. This rather strong statement is not really justified by the discussion in the manuscript, in my view. I therefore suggest that the authors explain more clearly what they mean (or maybe tone down their wording).

2. **Author Reply:** We understand the reviewer’s concern and while we believe the observed amplification effect is indeed a key piece of fundamental physics, we have followed the advice of the reviewer and we have removed this statement from the abstract.

We believe the reported effect will eventually open roads to the amplification of vacuum fluctuations by extracting energy from rotating bodies and hence despite being an effect of fundamental physics, it is also remarkably important for applications. Our paper unveils that it is actually possible to see and test the Zel’dovich effect with EM waves opening the road towards novel applications. We have explained our point more extensively in the manuscript, especially in the introduction, discussion and conclusions, where we also put the Zel’dovich effect into context of the wide literature on “quantum friction” effects, where one might expect the spontaneous generation of waves and the slowing down of the cylinder. An example of these effects is the dynamical Casimir effect, of which we have explained both similarities and differences with respect to Zel’dovich’s. Our paper is obviously classical and a quantum realization will still be very challenging, and might not be observed in the future, but we believe our proposal fosters research in that direction. See red text modification in the manuscript.

- Comment 2

1. **Reviewer Comment:** The authors should carefully check their Supplemental Material for consistency, clarity, and correctness. For instance, the LHS of (3) needs a prime, ω_{\pm} in (6) is not defined until the middle of page 3, the ϕ on the 2nd line of page 3 should probably be a Φ , what is σ in (9)?, the $f()$ in (15) is not clearly defined, the relation of F in (14),(15) to $F_{co}, F_{counter}$ is not clear, the F_{co} in (19) should probably be a $F_{counter}$, etc.

2. **Author Reply:** We thank the reviewer for this comment. We have fixed all the typos and added missing information to the supplementary file. All changes are in red text.

Response to Reviewer #2– NCOMMS-23-60951-T

We would like to thank the reviewer for taking the time to review the manuscript, and for giving constructive feedback. We are pleased that the reviewer is in principle agreeing to publish our paper in Nature Communications and we address their suggestions for improving our paper one-by-one below.

We have responded to all the comments in detail below, and therefore hope the reviewer now fully agrees that the paper in the improved version is in a form to be accepted for publication in Nature Communications. We have attached a copy of the manuscript with changes highlighted in red. Additionally, some wording in the manuscript has been tweaked to increase readability.

- Comment 1

1. **Reviewer Comment:** I would suggest that the authors produce a more tutorial description of the connection between their system and a rotating body amplifying some waves, as discussed by Zel'dovich. As it stands, the paper is highly technical, so only understandable by an intersection of two communities (electrical engineering and electromagnetic theory).
2. **Author Reply:** We thank the reviewer for this suggestion. In order to provide a clearer explanation of the connection between our experiment and Zel'dovich proposal and make the paper less technical, we have now rewritten the introduction to a large extent to explain the observed effect and its context better. Furthermore, we have also enlarged Fig. 1 adding the new Fig. 1a & 1b sketches that help us illustrate in a simpler and intuitive way the observed effect and its similarities to Zel'dovich's. Fig. 1a is a representation of Zel'dovich's original proposal - it shows a donut-shaped wave with orbital angular momentum (OAM) interacting with the spinning cylinder. In Fig. 1b we show how the previous illustration changes in our proposal: here the EM field with spin (polarisation) angular momentum (not OAM) impinges on the rotating body. The coloured arrows represent the (co and counter-rotating) spin components that arise from the decomposition of linearly polarized magnetic field B_0 incident on the cylinder. These sketches allow us also to show the similarities between the two configurations: both have a field with angular momentum impinging on the rotating body, so in both cases a negative Doppler-shift can be achieved. Also our geometry, not having the characteristic donut-hole of OAM waves allows us to increase the interaction between the field and the rotating body. For further clarity, we have also modified the text related to the circuit. Moreover, in the discussion section, we have explained how the Zel'dovich effect is related to other possible EM effects such as the dynamical Casimir effect, explaining the differences but also envisaging that our model could open novel roads to the observation of those effects as well.

- Comment 2

1. **Reviewer Comment:** I would suggest the authors discuss in more detail which aspects of the original prediction are covered here, as well as a better discussion of the different considerations that have been considered in the literature. The list of references misses important contributions in the field that can transmit an impression biasing.
In particular, a rotating body in a vacuum in which light waves are far from any sources constitutes an appealing class of amplification and spontaneous emission systems driven by mechanical rotation. These aspects have been covered in the following papers that should be appropriately cited in the manuscript:
 - Manjavacas and Garcia de Abajo, Phys. Rev. Lett. 105, 113601 (2010)
 - Asenjo-Garcia et al., Phys. Rev. Lett. 106, 213601 (2011)
2. **Author Reply:** We agree with the reviewer that there are other possible mechanisms leading to quantum friction and spontaneous generation of radiation from rotating bodies.

In particular, we can make a clear distinction between Zel'dovich effect and other effects related to the dynamical Casimir effect. The latter predicts emission at $1/2$ of the rotation frequency which is different from our observation. We provide a detailed overview of the relationship between other quantum friction effects such as the dynamical Casimir effect in the last paragraph of discussion. We actually then use this comparison to also explain in more detail the main features of the Zel'dovich effect. We have included some references to support the connection of our result to existing literature.

- Comment 3

1. **Reviewer Comment:** The spontaneous generation of radiation should occur for any rotation frequency in objects capable of absorbing at low frequencies (for example, dissipative metals). The process is associated with a simultaneous excitation in the material, emission of a photon, and change in the rotation state. The description in the manuscript of this point seems to be confusing and does not emphasize that this is a mechanism taking place even rotationally symmetric bodies.
2. **Author Reply:** We thank the referee for pointing out the need for more context. As explained in the previous comment reply, we have now added, in the last paragraph of the Discussion, a comparison with other known effects. We use this explicit comparison to our work to highlight the main features of both the dynamical Casimir effect (that has no frequency threshold) and the Zel'dovich effect (that has a frequency threshold).

- Comment 4

1. **Reviewer Comment:** The authors claim that it is impossible to observe amplification with light ("impossibility to test Zel'dovich predictions"), but some possible systems have been discussed in the literature including the Asenjo-Garcia et al. paper.
2. **Author Reply:** Yes, we stand by that claim although of course, one should probably refrain from ever saying that something is impossible. The effects described by Asenjo-Garcia et al. are of a different nature. There are of course similarities but the details are important and the two amplification effects arise from different physics. This is now explained in the revised text, just before the conclusions where we now also cite these papers. We should also underline that there was nothing in Zel'dovich's original proposal that explicitly prohibits observing the effect with light. Rather, it is simply a technological feasibility issue. Likewise, we have no doubts that it is possible to devise other theoretically possible scenarios where light amplification is possible. Our claim in this work is not that such proposals are wrong or cannot exist but rather, that we have provided the first experimental evidence where the Zel'dovich effect can be observed with EM waves. We have removed any mention regarding the impossibility of these effects being demonstrated also with light.

- Comment 5

1. **Reviewer Comment:** The conclusions in the paper seem to transmit the impression that the present work closes the subject by demonstrated something that is possible (and that can be further studied in analogous systems), and that any other configuration (particle in vacuum) is impractical.
2. **Author Reply:** We thank the reviewer for making us aware of this possible reading of our conclusion. This is of course not our intention and we believe there are other ways to demonstrate the effect. We only claim ours is the first experiment and only *one* possible way to do this experiments. We have now rephrased our statements throughout the manuscript in this regard. We have also added a sentence in the conclusions: 'Similarly, we expect that some of the approaches used here, e.g. resorting to low-frequency EM waves and the near-field configuration enabled by the interaction based on spin angular momentum, will be of use also for the further development and experimental implementation of other mechanical-to-EM wave transduction and amplification schemes.'

- Comment 6

1. **Reviewer Comment:** The similarities and differences with respect to the original proposal, and the range of possibilities should be discussed in a more balanced way.
2. **Author Reply:** As already pointed out in the reply to Comment 1 of the reviewer, we have enlarged the explanation of the similarities and differences between our proposal and Zel'dovich's original one. For this reason, we have largely rewritten the introduction and we have also added the Fig. 1a & 1b to help us illustrate this point.

Response to Reviewer #3– NCOMMS-23-60951-T

We would like to thank the reviewer for taking the time to review the manuscript, and for giving constructive feedback.

We have responded to all the comments in detail below, and therefore hope the reviewer now agrees that the paper is in a form to be accepted for publication in Nature Communications. We have attached a copy of the manuscript with changes highlighted in red. Additionally, some wording in the manuscript has been tweaked to increase readability.

- Comment 1

1. **Reviewer Comment:** After going through the work, I cannot recommend its publication in NC, due to the lack of novelty in physics and tight limitations in applications. Also the amount of work of this paper is far short of the bar of NC.
2. **Author Reply:** We thank the reviewer for sharing their opinion, but respectfully disagree.

Novelty in physics: Our experiment is the first clear experimental demonstration with EM fields of the Zel'dovich effect. There have been many experimental proposals since the initial work by Zel'dovich but no experiment, and the experimental testability and even the existence of the effect has been debated since 1971. Following the logic of the reviewer would mean to not value the contribution of experiment to the advancement of science. We believe that our experiment is a significant step forward as it gives confidence and confirmation in that this EM amplification does exist at all.

Applications: We believe the reported effect will eventually open roads in the future to the amplification of vacuum fluctuations by extracting energy from rotating bodies and hence despite being an effect of fundamental physics, it is also remarkably important for applications. Our paper unveils that it is actually possible to see and test the Zel'dovich effect with EM waves opening the road towards novel applications. We have explained our point more extensively in the manuscript, especially in the introduction, discussion and conclusions, where we also put the Zel'dovich effect into context of the wide literature on “quantum friction” effects, where one might expect the spontaneous generation of waves and the slowing down of the cylinder. An example of these effects is the dynamical Casimir effect, of which we have explained both similarities and differences with respect to Zel'dovich's. Please see red text modification in the manuscript.

Data: We have added more data to the paper, see the new Fig. 4, which shows the recorded amplitude data of the EM voltage for different frequencies of the rotor. This allows us to show directly the amplification if the Zel'dovich condition is reached.

- Comment 2

1. **Reviewer Comment:** The physics was actually pointed out in many previous works, such as Refs 11-13. The difficulty in realizing the amplification effect is mainly technical. For example, it is very challenging to rotate a cylinder super fast up to the frequency of light in THz or even sound in the kHz. In this work, the authors propose a LC circuit to generate radio waves with a very low frequency of hundreds of Hz which smartly relieve the tight requirement of rotation speed of cylinder. Such proposal cannot be easily extended into other fields such as acoustics and photonics.
2. **Author Reply:** We agree with the reviewer that the physics of Zel'dovich effect was pointed in previous works and that the difficulty is mainly technical, although it should be noted that some limitations are of fundamental character. For instance no rotating objects can be spun with tangential velocity higher than speed of light. In fact material breaking limits are achieved well below this threshold, at a tangential speed comparable to the speed of sound. These difficulties are explained in the introduction, which has been now further extended to take into account also the comments by another reviewer.

We also note that the reviewer acknowledges that our approach *smartly relieves the tight requirement of rotation speed*. Indeed, our claim is that this is the first experimental evidence of Zel'dovich prediction, albeit in a low frequency setup. The tight field confinement achievable in a lumped inductor in a deep near field regime is the key ingredient that allowed us to overcome the technical difficulties pointed out above.

We respectfully disagree with the critical remark of the reviewer that *this proposal cannot be extended into other fields*. We firmly believe that the same approach could relieve the Zel'dovich requirements in different situations. In fact, an experimental observation of Zel'dovich effect in acoustics has been already published in literature (Ref. 13, discussed in the introduction) and notably it shares similarity with our approach, in that the experiment is realized in a near field regime. Furthermore, we have already proposed an electromagnetic version of our experiment (Ref. 16) based on propagating electromagnetic waves confined in a waveguide. This experiment, although technically challenging, is in principle within reach with demonstrated technologies, and will allow extension in the realm of propagating waves. The same concept could be in principle extended to photonic structures. Indeed, one of the other referees asked us to explicitly indicate in the manuscript that these approaches cannot be extended to light.

- Comment 3

1. **Reviewer Comment:** This work has nothing to do with the quantum regime but only a LC circuit that generates radio waves of very low frequency and interacts with a spinning cylinder.
2. **Author Reply:** We agree with the statement of the reviewer as we do not claim ours is an experiment in the quantum regime. We describe our classical experiment to test and indeed experimentally confirm the Zel'dovich amplification effect – which is by using an LC circuit that generates a low frequency oscillating magnetic field and interacts with a spinning cylinder – and then give a perspective for how to perform a similar EM experiment in the quantum regime demonstrating the same effect based on our platform. We conclude that this prospect for a future quantum experiment is feasible. We hope this response does clarify our view on this remark for the reviewer. We also clarified in multiple points in the manuscript that we are dealing with the classical regime.

Response to Reviewer #4– NCOMMS-23-60951-T

We would like to thank the reviewer for taking the time to review the manuscript, and for giving constructive feedback.

We have responded to all the comments in detail below, and therefore hope the reviewer now agrees that the paper is in a form to be accepted for publication in Nature Communications. We have attached a copy of the manuscript with changes highlighted in red. Additionally, some wording in the manuscript has been tweaked to increase readability.

- Comment 1

1. **Reviewer Comment:** Authors reports an experimental method to observe the Zel'dovich prediction of amplification of electromagnetic wave by an induction motor with a spinning metal rod. The topic is very interesting and potentially provides a novel way to verify a novel interaction between the mechanical spin and electromagnetic (EM) wave. The measuring method the authors figured out is very clever. However, as for me, what the authors measured is only the negative resistance in the rotator-LC resonator, but not the actual gain or amplification of EM wave. The measured negative resistance just confirms the fact that the mechanical energy from the spinning object can be transformed to EM energy, but not to EM wave as the authors claimed in the manuscript.

Although the work is interesting, the physical process beneath the energy transform from the spin mechanics to EMs cannot be directly measured by the experiment. Thus, the “amplification of EM waves” in the title cannot accurately describe the actual work in the manuscript and may mislead reader. To confirm the conclusion more firmly and more rigorously, authors should present more measurements on the radiation of the EM wave when the spin speed is beyond the resonant frequency, confirming the amplification of the EM wave?

2. **Author Reply:** We thank the reviewer for expressing our approach as being ‘*novel*’ and ‘*clever*’. We agree with the reviewer that the amplification of EM wave/field was not shown directly in the previous version of the paper as we took the link to the data shown in Fig. 3 for granted. We have now included additional data and a new Figure (Fig. 4) showing directly the amplification effect of the electromagnetic field energy by an increase in amplitude, if the Zel'dovich amplification condition is met. Indeed, the new Fig. 4 directly shows that when the cylinder rotation rate is low the peak amplitude of the induced voltage (at resonance frequency of the circuit) is lower than that of the no-rotor case. This is due to the presence of eddy currents that make the field dissipate inside the cylinder. Once though the rotor rotation is increased to higher values (and Zel'dovich condition is satisfied), we see that the peak amplitude also increases and eventually, exceeds the no-rotor peak amplitude. This represents a direct measurement of the back-reflected field from the cylinder, hence provides a direct measurement of the EM field amplification. We hope the reviewer agrees this clarifies this point.

We have also changed the manuscript title into ‘...amplification of electromagnetic fields ...’ to clarify what is evidenced by our data.

- Comment 2

1. **Reviewer Comment:** My other concern is about the style of writing the manuscript. In the manuscript, the main novel idea beneath the experimental method is somewhat abstract and physical to make some difficulty for readers. Could authors give some figures to explain the idea of the experiment and its connection to the Zel'dovich prediction?
2. **Author Reply:** We thank the reviewer for making this point and have now rewritten the introduction and also the discussions section to put the Zel'dovich amplification into context of somewhat related effects such as the dynamical Casimir effect and how our experiment is testing the prediction by Zel'dovich. We have added Fig. 1a & 1b,

including description in the manuscript text and the figure caption to illustrate the amplification effect and explain the idea of the experiment. Figure 1a is a representation of Zel'dovich original proposal - it shows the donut-shaped wave with orbital angular momentum (OAM) interacting with the spinning cylinder. In Fig. 1b we show how the previous illustration changes in our proposal: here the EM field with spin (polarisation) angular momentum (not OAM) impinges on the rotating body. The coloured arrows represent the (co and counter-rotating) spin components that arise from the decomposition of linearly polarized magnetic field B_0 incident on the cylinder. These sketches allow us also to show the similarities between the two configurations: both have a field with angular momentum impinging on the rotating body, so in both cases a negative Doppler-shift can be achieved. Also our geometry, not having the characteristic donut-hole of OAM waves allows us to increase the interaction between the field and the rotating body. Please see changes in red colour in the manuscript.

- Comment 3

1. **Reviewer Comment:** My third concern is about the claim in the manuscript that the experiment could observe EM amplification in quantum regime. It is well known that the noise induced by quantum fluctuation in vacuum EM fluctuation may be far small than the noise in electronic system as used in the experiment even in the ultra-low environmental temperature. So, I think the claim is not rigorous and the EM amplification in quantum regime may not be observed.
 2. **Author Reply:** We agree with the referee that extending these experiments to the quantum regime will be challenging. We explicitly comment on that in the conclusions. However, such an experiment in the quantum regime is clearly future work and will be remarkably challenging. But we still believe that this work is a significant step forward in the right direction. The first step has to be the observation of "classical" amplification. This has been achieved in this work. Now future endeavours need to focus on improving the efficiency and frequency range in which this can be operated. Indeed, we agree that the quantum version of the Zel'dovich effect will be challenging as one needs to suppress by many orders of magnitude thermal/electronic fluctuations in the circuit. However, we argue that with our platform, there is in principle no fundamental show stopper to achieve also such quantum effect. In particular, in the conclusion we mention a concrete approach to overcome the above issue (feedback-cooling). There is a chance that the EM amplification in the quantum effect may not be observed, but we believe that our platform is the strongest contender amongst experiments to try to observe it instead.
-

REVIEWER COMMENTS

Reviewer #2 (Remarks to the Author):

The authors have substantially improved their manuscript and comprehensively responded to all criticisms by the referees in a convincing way.

Reviewer #3 (Remarks to the Author):

After going through this work, I find that the manuscript has been improved by adding some figures and descriptions on the physics. My concern is still on the following points:

1. This work cannot be easily extended to acoustics and optics due to the high-frequency characteristics.
2. The demonstration data is weak. I expect more effects caused by the amplification of EM such as turn on a LED light (make it much more brighter) due to the amplification.

The physics is not that novel and the application potential is not clear without some demonstrations. The workload inside this work is not enough for nature communications.

Reviewer #4 (Remarks to the Author):

Part of my concerns are addressed by authors. However, there are still some concerns about the manuscript.

- (1) Although authors insert Fig. 1 (a) and (b) to describe the scheme of interaction between OAM (SAM) and spin metal cylinder, the figures do not show the critical idea of “amplification”. So, I think the figures should be improved to include more information about the main idea of the work.
- (2) According to the Zel’dovich prediction, the absolute value of the negative resistance induced by rotational frequency should increase with the rotational frequency of the metal cylinder, and thus leading to larger value of the negative resistance at higher rotational frequency. However, in Fig. 3 (a) and (c), it is clearly seen that the induced negative resistance will tend to zero with the increase in the rotational frequency F . There is somewhat a conflict to that. Could authors give some comments to explain that?
- (3) Additionally, since the amplification condition $\omega - \Omega < 0$ is a critical threshold from absorption to amplification for the system. The Figs. 3 (a-d) does not show such a transition point, since, as for me, the transition should change suddenly, but the resistance just decrease slowly (not suddenly) around resonance frequency. Could author explain the phenomenon?
- (4) There is a strange phenomenon that is needed to be addressed in the Figs. 3(a-d) and (e-h). According to the amplification condition $\omega - \Omega < 0$, the resistance induced by the rotational frequency Ω should only decrease when $\Omega > \omega$ resonant frequency. However, the Figs. 3 show the decrease in the induced resistance appears in the region of $\Omega < \omega$. That is not consistency with the amplification condition. Since that is critical support for the work, authors should present deep analysis and give comments on it.

(5) Since authors extends the discussions about the connection to quantum vacuum, could authors give some measurements around the resonant frequency without external voltage signal, i.e. with removing the V_i in Fig. 1 (d) ?

So, I think the current form of the manuscript is not good enough for publication on NC.

Response to Reviewer #2– NCOMMS-23-60951-T

We would like to thank the reviewer for taking the time to review our revised manuscript, as well as for the previous suggestions that helped us to communicate the research in a clearer way. We are pleased that the reviewer continues to support the publication of the paper in Nature Communications and believes the revised manuscript is now much stronger for the improvements made, that allowed us to address all the previous criticisms convincingly.

Response to Reviewer #3– NCOMMS-23-60951-T

We would like to thank the reviewer for taking the time to review our revised manuscript, and for the approval of the improvements. We regret that this has not been sufficient to change the reviewer's views on the value and significance of the research presented in the paper.

We have responded to the comments in detail below. We have attached a copy of the manuscript with changes highlighted in red.

- Comment 1

1. **Reviewer Comment:** After going through this work, I find that the manuscript has been improved by adding some figures and descriptions on the physics. My concern is still on the following points:

1. This work cannot be easily extended to acoustics and optics due to the high-frequency characteristics.

2. **Author Reply:** We believe there is a misunderstanding on this matter. We kindly disagree with the impossibility to extend this work easily to acoustics. Indeed, many of the authors have already shown the presence of the Zel'dovich amplification effect with acoustics, and at low frequency. Please refer to Ref. 13 in the paper. Further application to acoustics is not a concern of this paper, which sets out to prove that the Zel'dovich effect can be seen in a different system - i.e. in electromagnetism. We agree that the Zel'dovich effect is likely technically very difficult to observe in optics (and high frequency waves), that is why we have taken this approach at low frequencies, it being an actually accessible regime to provide the first proof in electromagnetism. We maintain that this is a significant step into research of this fundamental effect.

- Comment 2

1. **Reviewer Comment:** 2. The demonstration data is weak. I expect more effects caused by the amplification of EM such as turn on a LED light (make it much more brighter) due to the amplification.

2. **Author Reply:** We regret we do not see the relevance of proving the amplification through an LED increased brightness. We believe that it is better in a scientific paper to prove amplification with a direct measurement of the signal having increased. We believe this is sufficient and clearest rigorous scientific evidence of the effect. We have provided this in the previously added Fig. 4 (now Fig. 5), which explicitly shows how the signal corresponding to the current in the circuit (and so in turn to the total magnetic field in the toroid gap) is amplified once the cylinder rotation speed exceeds the Zel'dovich condition. In Fig. 4 (now 5), the increase in amplitude of the EM voltage is the amplification: the LED would be brightest then the no rotor case when the cylinder would rotate at $F = 330 - 400$ Hz (violet and cyan curve) and the EM frequency be $f \sim 277$ Hz. In the old Fig. 3 (now Fig. 4) we also show how the measurements with many different resonant circuits match to the theoretical model of the Zel'dovich effect.

- Further comments

1. **Reviewer Comment:** The physics is not that novel and the application potential is not clear without some demonstrations. The workload inside this work is not enough for nature communications.

2. **Author Reply:** We respectfully disagree with the reviewer also on this point. Despite it is true that the physics is not novel since this effect was postulated by Zel'dovich in 1971, we are demonstrating for the first time after over 60 years that this long-sought effect is measurable also for electromagnetic waves. For acoustic waves, as said before, it had been demonstrated only a few years back (See again Ref. [13]). So the new physics relies in proving the effect for electromagnetic fields. Furthermore, potential applications

of this research are actually very clear. This is one of those cases where the application came before the actual understanding of the physics. Indeed, the analogue of this effect has been engineered in the best way possible and it is used in the present technology of windmills. Induction generators are widely used in engineering and energy systems and this manuscript offers a new perspective on their technology, one so essential for human existence. So we firmly disagree with the reviewer on this point too.

We would also like to comment on the matter that “the workload is not enough”. We strongly believe that the amount of workload should not factor in as a scale of scientific significance. Some discoveries can take longer than others, but the amount of time should not weight on the importance of the discovery. We understand that the experiment looks simple, however the simplicity of it does not cancel the step forward that it takes us in knowledge and understanding of this long-sought phenomenon. We believe that what counts is the scientific importance and the evidences provided to support the claim. In this regard, we provided all the measurements needed to demonstrate unequivocally the presence of the effect. When asked for, we provided new measurements that confirmed the previous findings. We have proven the persistence of the effect for many frequencies and also provided the analytical modeling of the experiment, which is in good agreement with the experimental results. These results hence provide the first measurement of the Zel’dovich amplification for EM wave and a conceptual step forward in the understanding of the physics of this 60 year-old phenomenon, whose importance is not only merely epistemic, but also pragmatic since only once classical amplification is fully understood and proven, one can start pursuing the quantum effect, which is quite distant yet still on the horizon.

Response to Reviewer #4– NCOMMS-23-60951-T

We would like to thank the reviewer for taking the time to review our revised manuscript, giving feedback on the changes and for raising areas where they still have concerns. We hope that our response clears up these issues, and we have also made changes to the manuscript by editing and adding figures and additional explanation to the manuscript and supplementary to clarify similar issues for readers.

We have responded to all the comments in detail below, and therefore hope the reviewer now agrees that the paper is in a form to be accepted for publication in Nature Communications. We have attached a copy of the manuscript with changes highlighted in red.

• Comment 1

1. **Reviewer Comment:** (1) Although authors insert Fig. 1 (a) and (b) to describe the scheme of interaction between OAM (SAM) and spin metal cylinder, the figures do not show the critical idea of “amplification”. So, I think the figures should be improved to include more information about the main idea of the work.
2. **Author Reply:** We thank the reviewer for the suggestion. Indeed, for someone just glancing through the figures before reading the paper, while Fig. 4 (now Fig. 5) directly shows the amplification measurement of our signal, there was no emblematic visual that represented the broad concept of how the amplification manifests in our setup. We have now edited Fig. 1c to add a figurative diagram. This shows the amplification (or absorption) of the magnetic field generated by the coil and toroid, due to the presence of the rotating cylinder, and making it also clear here that the amplification is with respect to the situation with no cylinder in the gap.

• Comment 2

1. **Reviewer Comment:** (2) According to the Zel’dovich prediction, the absolute value of the negative resistance induced by rotational frequency should increase with the rotational frequency of the metal cylinder, and thus leading to larger value of the negative resistance at higher rotational frequency. However, in Fig. 3 (a) and (c), it is clearly seen that the induced negative resistance will tend to zero with the increase in the rotational frequency F . There is somewhat a conflict to that. Could authors give some comments to explain that?
2. **Author Reply:** We thank the Reviewer for this comment, which gives us the opportunity to clarify the possible misconception when thinking of the Zel’dovich amplification and in particular of the amplification condition, Eq.(1) of the manuscript: the misconception being that the Zel’dovich effect predicts an always decreasing resistance with increased cylinder frequency, i.e. an always increasing amplification with cylinder rotation frequency. The Zel’dovich effect relies on the fact that the absorption response of a rotating object is odd-symmetric (i.e. flips to amplification) where negative frequencies are present. The details of the shape of that absorption response function are system-dependent, and are more complicated than a monotonically decreasing absorption (resistance). Indeed, here for a conductive cylinder interacting with an EM field, the negative resistance is predicted to slowly reduce in magnitude (tend to zero) with additional rotation speed after reaching a minima (maximum gain - see old Fig. 3, new Fig. 4 in the manuscript). This behaviour is already present in Zel’dovich’s original papers - see in Ref. [3] considerations after Eq. (2.5)).

We can provide an intuitive physical picture for this behaviour. The magnitude of the field interaction with the cylinder is associated with the response function of the cylinder to the incident EM field. The dissipation/amplification of power from the EM field into the metallic cylinder is due to eddy currents, and is described mathematically by

the dissipative part of the response function (which we call χ'' in the manuscript) of the rotating cylinder to the EM field. In particular then, the Zel'dovich effect depends on the shape of this dissipative part χ'' , which is odd-symmetric: at negative frequencies the dissipative response function flips sign and the field dissipation (absorption) becomes gain (amplification) - the equivalent amount of power that was previously absorbed is now emitted from the cylinder. χ'' in normal metal is characterized by a length scale, usually called penetration depth, $\delta(\omega) = (\sigma\mu_0\mu_r\omega)^{-1/2}$, where σ is the conductivity, μ_0 and μ_r are the vacuum and relative permeability, respectively. For a rotating cylinder we need to replace ω with $\omega_{\pm} = \omega \pm \Omega$ for the corotating and counterrotating components present in our experiment (although only the co-rotating component ω_- reaches a negative frequency). The magnitude of the dissipation $|\chi''|$ (see the new Fig. S1 in the supplementary reported here as Fig. 1 below) is maximized when the penetration depth approaches the system size (characterised by our cylinder radius a). This means that there is a maximum absorption (positive χ'') and a maximum amplification (negative χ'') when $|\omega_-|$ is such that the penetration depth is of the order of the cylinder radius a (we are referring to the corotating frequency ω_- being the only one to undergo negative Doppler-shift when $\Omega > \omega$, in our experiment).

FIG. 1. Dissipative response of the sphere χ'' , normalized on the factor $2\pi a^3/\mu_0$ as a function of the normalized field frequency in the corotating frame ω_-/ω_* .

For our cylinder we estimate that the magnitude of the dissipation $|\chi''|$ is maximized for $|\omega_-| = \omega_m \approx 2\pi \times 97$ Hz. To calculate this we defined $\omega_* = (\mu_0\mu_r\sigma a^2)^{-1}$ as the scaling frequency at which the penetration depth equals the radius a of the sphere (we are using the sphere model to compute this (see SM file for more details)) and then we observed that $|\chi''|$ is maximum at $\omega_m = 11.6\omega_*$ ($\omega_* \approx 2\pi \times 8.4$ Hz in our setup) - see Fig. 1 (Fig. S1 in the supplementary file). This defines the width of the transitions of resistance from positive to negative values. Indeed, looking at Fig. 4 of the manuscript, the maximum gain (i.e. the minimum of the resistance curves) is always located at $\omega_m \approx 2\pi 97$ Hz from the flipping frequency (where $\omega = \Omega$ specified by the vertical dashed line). In the limits, $|\omega_-| \ll \omega_m$ or $|\omega_-| \gg \omega_m$ the dissipative response function χ'' goes to 0. The physical picture in the two cases is as follows. For $\omega_- \rightarrow 0$ there is full penetration and negligible induced eddy currents, the field mostly passes through without being absorbed. For $\omega_- \rightarrow \infty$ eddy currents flow only in a thin layer of metal and most of the incident field is reflected without being absorbed. Note that these considerations of penetration depth were already clearly stated by Zel'dovich, see for instance Ref. 3 (specifically, Section 2 refers to penetration depth, equations 2.4 and 2.5 give the limits for a similar case). We hope we have now adequately explained to the Reviewer how our results have no conflict with Zel'dovich's prediction.

Since this physical insight was not properly discussed in the previous version of the

manuscript, we added a paragraph in the model section of main paper with a short intuitive explanation of the connection of the dissipation behaviour as function of frequency to the penetration depth (see red text). We have also added Figure 2 in the main paper showing this response, and how the co- and counter-rotating terms add (for this see also two cases reported in Fig. 2 here - Fig.S2 of the Supplementary file). For the interested reader we have added a longer and more detailed discussion in the Supplementary Material, which connects the above considerations to the mathematical modeling. We have also added three Supplementary figures showing the universal behaviour of the dissipation of a conductive rotor as a function of the normalized frequency, where the normalization takes into account the skin penetration depth. With this, we hope to have addressed most of the issues raised by the reviewer.

FIG. 2. (a): Dissipative response of the sphere χ'' as function of normalized rotation frequency for the the two components corotating (blue line) and counterrotating (cyan line). The red line is the sum of the two terms, showing that the total dissipation can indeed become negative. Here the field frequency is set to $\omega = 50\omega_*$. (b): the same but now for $\omega = 500\omega_*$.

- Comment 3

1. **Reviewer Comment:** Additionally, since the amplification condition $\omega - \Omega < 0$ is a critical threshold from absorption to amplification for the system. The Figs. 3 (a-d) does not show such a transition point, since, as for me, the transition should change suddenly, but the resistance just decrease slowly (not suddenly) around resonance frequency. Could author explain the phenomenon?
2. **Author Reply:** Yes, we are happy to explain this point further. ‘Slowly’ and ‘suddenly’ are relative terms. As discussed in the previous point response the transition from positive to negative values of resistance is predicted to occur over a frequency width set by the penetration depth of the electromagnetic field. In our experiment due to the size a of our cylinder, the transition from maximum absorption to maximum amplification occurs

within $\omega_m \approx 2\pi \times 97$ Hz of the critical threshold frequency. In the Fig. 3 (now Fig. 4) graphs at higher circuit resonance frequencies, the transition appears sharper, due to the enhancement effect of higher frequencies (including the ω prefactor in Eq. 4 in the main text, and the asymmetry of the two components), but the transition width is always the same.

The critical threshold transition point is not a maximum (or minimum) in the theoretical Zel'dovich Ω -resistance curve but an inflection point. See Fig. 1 and 2 in this response letter, Fig. 2 in the main paper and new figures in the supplementary material. In Figs. 3 (a-d) of the main text that show our results it can be seen that when the cylinder is at the resonance frequency (the marked vertical lines) it indeed coincides with the inflection point of the data - and this *is* also the place in the graphs where the resistance is changing most rapidly with cylinder rotation frequency. Our data measure the resistance of the cylinder to the counter and co-rotating components combined, and so this inflection point does not sit at zero cylinder resistance (as it would for the co-rotating only case), but above it, due to the counter-rotating term (ω_+) that can never reach a negative frequency and so does not amplify (the cyan line in the response letter Fig. 2). Nevertheless, because the counter rotating term is approximately linear in the transition region, the inflection point still coincides with $\omega - \Omega = 0$ (when the rotation matches the resonant circuit frequency) due to the presence of the co-rotating term changing curvature and switching to amplification when ω_- goes negative.

We have made changes to the manuscript and the supplementary, referred to in the response to comment 2, detailing the physics of the penetration depth and the transition width which clarify this point for readers.

- Comment 4

1. **Reviewer Comment:** (4) There is a strange phenomenon that is needed to be addressed in the Figs. 3(a-d) and (e-h). According to the amplification condition $\omega - \Omega < 0$, the resistance induced by the rotational frequency Ω should only decrease when $\Omega > \omega$ resonant frequency. However, the Figs. 3 show the decrease in the induced resistance appears in the region of $\Omega < \omega$. That is not consistent with the amplification condition. Since that is critical support for the work, authors should present deep analysis and give comments on it.
2. **Author Reply:** We understand the concern, however our data are consistent with the Zel'dovich amplification condition, which states that the absorption of a rotating cylinder in a field with co-rotating spin becomes negative when $\omega - \Omega < 0$. The key point here is that amplification or absorption is defined with respect to the field when the cylinder rotor is not present at all, *not* with respect to a case where the cylinder is present. The cylinder's absorption of (resistance to) the co-rotating field has to decrease before the $\Omega = \omega$ point, to be able to go to zero at that point. The zero absorption case at $\omega_- = 0$ is equal to the case where the cylinder rotor is not present at all - in the rotating frame the cylinder 'sees' the co-rotating field as a field of zero frequency, for the co-rotating field it is like the cylinder is not there at all, and its resistance has gone to zero. At cylinder speeds greater than this, the absorption of the co-rotating field instead becomes amplification (the resistance of the cylinder to the co-rotating component decreases further, so below zero and becomes negative). One remark though has to be done at this point. One could ask why then in Fig. 5 of the manuscript the curves at $F = 290$ Hz is below the one with no rotor. We have to remember that in our experiment there is always the counter-rotating component adding an additional absorption with respect to the no-rotor case. This makes the maximum of the curve at $F = 290$ Hz lower than the no-rotor case. A rotating field would be needed in order to cancel that contribution and see the cylinder 'disappear' when $\omega = \Omega$. We have addressed the issue of the specific shape of the Zel'dovich predicted absorption-amplification curve in the points above, and again

stress these key concepts can all be found in Zel'dovich's original papers, even if the implementation is slightly different.

The changes to the manuscript and supplementary explaining the shape of the response function with respect to the physics of penetration depth in response to the previous points should clear up confusion around this point as well. Furthermore, with the improved Fig. 1c, we have made it clearer that the loss or gain is measured with respect to the no cylinder case.

- Comment 5

1. **Reviewer Comment:** Since authors extends the discussions about the connection to quantum vacuum, could authors give some measurements around the resonant frequency without external voltage signal, i.e. with removing the V_i in Fig. 1 (d) ?
 2. **Author Reply:** This is indeed a very interesting question. Unfortunately though, if we remove the input voltage in this setup we see no signal above the noise floor of the detector. We have added a sentence to this effect in the 'experimental setup' section (red text). With only noise as input, the signal to noise is insufficient to properly measure the resonant response of the circuit, let alone any change due to the cylinder. The resistance in the total circuit $R_0 + \mathcal{R}$ (not just the effective resistance of the cylinder \mathcal{R}) is large, due to the $R_0 = 2.03 \text{ k}\Omega$ resistance of the coil and parallel dielectric losses. The negative resistance of the cylinder (which is also restricted by the presence of the counter-rotating polarisation component in our oscillating magnetic field) is a sizeable change to that total, but not large enough to create a total negative resistance in the whole circuit. Therefore, there is no observable spontaneous amplification in this experimental case; without a measurable input signal around the circuit resonance frequency (of which measurable oscillations die away near-instantaneously when the input is removed) amplification cannot be observed. However, with future experiments with different experimental apparatus (to create a rotating not oscillating field, increase the geometric coupling of the field/circuit to the cylinder, and reduce the circuit resistance) we hope to explore this no-input (only noise) regime. Also, as noted in Reviewer #4's previous comments, quantum vacuum noise is likely to be far smaller than electronic noise, even in the ultra-low temperature regime. Here we are at non-cryogenic temperatures and at these frequencies thermal noise vastly overpowers quantum vacuum noise, and this much larger thermal noise would be the seed for any signal with a set-up such as this, not quantum noise. We reassert that there is no observable effect of the quantum vacuum in this experimental setup. However, the reasons are only technical ones due to large resistances and the presence of additional stronger noise excitation sources, rather than conceptual ones, which is why we mention how this research, showing the Zel'dovich effect in electromagnetism, enables the possibility of scaling Zel'dovich experiments to the quantum regime.
-

REVIEWER COMMENTS

Reviewer #4 (Remarks to the Author):

Authors have addressed my concerns thoroughly. By now, I am glad to recommend the manuscript for publication. Additionally, since the EM amplification by a rotating rod is critical important, could authors provide more information of the experiment, such as Figures of experiment setup or videos of the highly rotating rod? So, the information will make it easier for other groups to recheck the results.

Response to Reviewer #4– NCOMMS-23-60951-T

We would like to thank the reviewer once more for taking the time to review our revised manuscript, and thank them for their recommendation to publish our paper.

We have responded to the comment in detail below. We have attached a copy of the manuscript/supplement with changes highlighted in red.

- Comment 1

1. **Reviewer Comment:** Authors have addressed my concerns thoroughly. By now, I am glad to recommend the manuscript for publication. Additionally, since the EM amplification by a rotating rod is critical important, could authors provide more information of the experiment, such as Figures of experiment setup or videos of the highly rotating rodSo the information will make it easier for other groups to recheck the results.
 2. **Author Reply:** We have now added photographs of the experimental setup and a brief description to the supplementary material.
-